# Impact of Dietary Protein Levels and Gender on Carcass Characteristics and Meat Quality in Slow-Growing Ducks

**DOI:** 10.3390/ani16010079

**Published:** 2025-12-26

**Authors:** Yong Jiang, Yijia Lu, Zhong Zhuang, Lei Wu, Yongpeng Li, Hao Bai, Yulin Bi, Zhixiu Wang, Shihao Chen, Guobin Chang

**Affiliations:** 1Key Laboratory for Animal Genetics & Molecular Breeding of Jiangsu Province, College of Animal Science and Technology, Yangzhou University, Yangzhou 225009, China; jiangyong12126@163.com (Y.J.); yijialu2023@163.com (Y.L.); zz150211@163.com (Z.Z.); 17320275567@163.com (L.W.); 17794281505@163.com (Y.L.); ylbi@yzu.edu.cn (Y.B.); wangzx@yzu.edu.cn (Z.W.); mrrchen@yzu.edu.cn (S.C.); 2Joint International Research Laboratory of Agriculture and Agri-Product Safety, Ministry of Education of China, Institutes of Agricultural Science and Technology Development, Yangzhou University, Yangzhou 225009, China; bhowen1027@yzu.edu.cn

**Keywords:** slow-growing duck, protein, fatty acid, amino acid, meat quality

## Abstract

Fast-growing ducks dominate meat duck production, but the associated high-density feeding leads to reduced meat quality and flavor. Slow-growing ducks with high levels of muscle fat, unsaturated fatty acids, and flavor compounds have better meat quality and are in greater demand. Meat quality is influenced by factors such as breed, sex, and diet, and a balanced diet is necessary for optimal growth and meat quality. However, there is limited information on the protein requirements of slow-growing ducks, and physiological differences exist in the nutritional requirements of male and female ducks. Therefore, it was hypothesized that the effect of protein on growth and meat quality of ducks may be sex-related, especially for slow-growing ducks. This study investigated the impact of different protein levels and sex on slow-growing ducks to reveal the relationship between dietary protein levels, sex, slaughter performance, meat quality, and muscle fatty acid and amino acid contents of slow-growing ducks.

## 1. Introduction

Fast-growing ducks dominate the duck meat production industry, with China’s slaughter volume of 4.2 billion accounting for 82.76% of the global duck meat industry in 2023, according to the Food and Agriculture Organization of the United Nations statistics. As is well known, fast-growing Pekin ducks can achieve a market weight of more than 3.0 kg at 5–6 weeks due to genomic selection and enhanced diet. However, the selection for intensive growth has led to adverse consequences, such as a decline in meat quality and flavor [1]. Meat quality is primarily influenced by tenderness and flavor [2]. Tenderness is determined by the characteristics of muscle fibers [3], whereas the taste and flavor of meat are influenced by intramuscular fat and amino acid contents [4,5]. By contrast, slow-growing ducks possess increased levels of intramuscular fat, unsaturated fatty acids (UFA), polyunsaturated fatty acids (PUFA), and ω-6 fatty acids, as well as increased ratios of PUFA/saturated fatty acids (SFA) and UFA/SFA [6,7,8]. They also possess elevated levels of flavor substances [9] and meat texture [10]. This indicates that slow-growing ducks have better meat quality. Consequently, there has been a recent increase in demand for slow-growing ducks in China, driven by the desire for high-quality meat.

Consumer preferences for meat are influenced by tenderness, color, odor, and taste, which are affected by breed, sex, age, dietary nutrients, and management practices [6]. Achieving optimal growth capacity and better meat quality requires a balanced diet. However, there is limited information regarding the protein requirements of slow-growing ducks. Slow-growing chickens have lower crude protein requirements than fast-growing chickens [11,12]. Protein, an essential element in livestock feed, is vital for the development of animals and overall meat quality. For instance, chickens fed a low-protein diet had dark meat and high drip loss, and an increased cross-sectional area of the breast muscle in male slow-growing chickens [13]. In ducks, a low-protein diet was found to increase the pH level and yellowness, while decreasing shear force and drop loss in the breast muscle of the duck [14]. In addition, female ducks had low protein content, and fiber cross-sectional area, and higher intramuscular fat content in breast and thigh muscle [15]. The nutrient requirements of male and female poultry exhibit distinct physiological profiles. Therefore, we speculated that the effects of protein on growth performance and meat quality in ducks may depend on the sex, especially in slow-growing ducks.

Therefore, the objective of the current experiment was to investigate the effect of dietary protein content and sex on carcass traits, body composition, meat quality, and concentrations of fatty acids and amino acids in slow-growing ducks.

## 2. Materials and Methods

### 2.1. Ethics Statement

All animal experimental procedures were approved by the Institutional Animal Care and Use Committee of the School of Animal Science and Technology, Yangzhou University (Permit Number: YZUDWSY, Government of Jiangsu Province, China).

### 2.2. Bird Management and Diets

A total of 6000 mixed-sex slow-growing ducklings were assigned to one of two dietary treatments at 1 day of age. Each treatment group included 6 replicates, with 500 ducklings per pen (10 × 10 m), and the average live weight was about 640 g in each group and each replicate. All ducklings were raised in enclosures padded with rice hulls. For the first 3 days, the temperature was maintained at 30 °C, then reduced to 25 °C until the ducks were 21 days old. From 1 to 21 days of age, all ducks received a granulated consistent diet. At 22 days of age, the ducks were fed either a low-protein or high-protein diet (Table 1). Throughout the experimental period, all the ducks had ad libitum access to food and water.

### 2.3. Sample Collections and Analyses

At 63 days of age, 100 ducks (50 male and 50 female) were selected randomly per pen and weighed after a 12 h fasting to measure weight gain, and four ducks (two males and two females) per pen (a total of 48 birds) were selected based on average body weight for slaughter carcass trait analysis.

### 2.4. Crude Protein and Amino Acids in Diet

Feed samples were finely pulverized to pass through a 0.50 mm mesh and mixed prior to analysis. Dietary protein levels were measured using the Kjeldahl method [16]. Dietary amino acid concentrations were determined through ion-exchange chromatography using an amino acid analyzer (L-800, Hitachi, Tokyo, Japan), following hydrolysis of diet samples in 6 mol/L HCl at 110 °C for 24 h.

### 2.5. pH and Color Profile in Breast and Thigh Muscles

The acidity levels of the breast and thigh muscles were measured from the same spot on the right breast and thigh muscle using a pH meter (pH-2004; Selecta, Barcelona, Spain), which was calibrated with calibration buffers of pH 4.01 and pH 6.86. Measurements were repeated three times on three different occasions per sample, following the methodology described by the previous method [17]. The flesh color attributes [lightness (L*), redness (a*), and yellowness (b*)] of the breast and thigh muscles were determined using a Minolta CR-400 colorimeter (Konica, Chiyoda-ku, Japan) with three measurements on the muscle surface after removing the outer membrane, following a 30 min exposure to ambient conditions [18]. Prior to analysis, the colorimeter was calibrated with the following parameters: D65 illuminator, Y = 88.8, x = 0.3178, y = 0.3350, viewing angle 2°, aperture diameter 8 mm.

### 2.6. Water Loss Rate of Breast and Thigh Muscles

The water loss rate was measured using a meat quality pressure meter (Meat-1, Tenovo Food, Beijing, China). Collected meat samples (0.125 cm^3^) were wrapped in absorbent paper and placed into the machine for testing. The program was set to 300 N for 5 min, and all samples were measured 3 times, with the results determined by averaging. The calculation method for the water loss ratio was as follows: weight of meat after pressing / weight of meat before pressing.

### 2.7. Chemical Composition of Breast and Thigh Muscles

The chemical composition was determined using the breast and thigh muscle surplus samples, with moisture, protein, intramuscular fat (IMF), and collagen in the meat and meat products determined using a near-infrared spectrophotometer with an artificial neural network calibration model and database within the FoodScan Meat Analyzer (FOSS FoodScan 78800; Dedicated Analytical Solutions, Hilleroed, Denmark). All exterior fat and connective tissue were removed prior to proximate analysis. Each sample was coarsely ground using a tabletop grinder to obtain a sample of approximately 180 g. The ground sample was then placed in a 140 mm round sample dish, and the dish was placed in a FoodScan. Protein, IMF, collagen, and moisture percentages (g/100 g) are shown. The final reported values were calculated by taking independent readings for each sample and averaging them. All measurements were carried out in triplicate.

### 2.8. Analysis of Fatty Acid Content in Breast and Thigh Muscles

Fatty acids were extracted from the breast and thigh muscles [19]. The extracted fatty acids were then methylated by adding 1 mL of acetyl chloride in methanol (1:10, *v*/*v*). To analyze the fatty acid profiles, 1 μL of the supernatant was injected into a 7890A GC-FD system from Agilent Technologies located in Palo Alto, CA, USA. Fatty acid separation was achieved by utilizing a DB-23 column (60 m × 0.25 mm × 0.25 μm) manufactured by Agilent Technologies with a split ratio of 1:50. The injector and detector were kept at 250 °C, while the oven was programmed as follows: starting at 50 °C for 2 min, gradually increasing to 175 °C at a rate of 20 °C/min, then to 220 °C at a rate of 2 °C/min, and finally to 230 °C at a rate of 4 °C/min. The temperature was maintained at a constant level for 5 min, with nitrogen used as the carrier gas at a flow rate of 1.1 mL/min.

### 2.9. Analysis of Amino Acid Content in Breast and Thigh Muscles

Amino acid profiles of the breast and thigh muscles were analyzed using an amino acid analyzer (LA8080; HITACHI, Tokyo, Japan). Approximately 100 mg of breast muscle tissue was ground into a fine powder. The powdered tissue was then transferred to a 15 mL glass hydrolysis bottle with a lid, followed by adding 10 mL of 6 mol/L HCl for acid hydrolysis, and then phenol was added slowly. The mixture was subjected to hydrolysis at a temperature of 110 °C for a duration of 22 h. Following hydrolysis, the hydrolysate was transferred to a 50 mL flask and diluted with ultrapure water. The resulting solution was filtered through a 0.22 μm membrane filter into a vial for the autosampler. Ion-exchange chromatography was employed to separate the different amino acids using lithium citrate buffer as the eluent. After post-column derivatization with ninhydrin, the derivatives were detected at wavelengths of 570 and 440 nm.

### 2.10. Statistical Analysis

We first tested the normal distribution of the data with the Shapiro–Wilk test in the SAS 9.4 software [20]. The data from the experiment were subjected to nested model ANOVA with the GLM procedure. The SNK’s method was used to test the difference between means. Statistical analysis was conducted with the significance level set at *p*-value < 0.05.

## 3. Results

### 3.1. Carcass Traits

Dietary protein levels had no influence on carcass traits of the slow-growing ducks (*p* > 0.05, Table 2). The carcass traits of the slow-growing ducks had no differences between female and male ducks, whether fed with high or low-protein diets (*p* > 0.05).

### 3.2. Meat Quality of Breast and Thigh Muscles

Dietary protein levels had no influence on pH1, pH24, lightness (L*), redness (a*), and yellowness (b*) in breast muscle and thigh muscle of the slow-growing ducks (*p* > 0.05; Table 3). Female ducks had higher pH1 of breast on high-protein diets, and higher pH24 of breast on low-protein diets (*p* < 0.05). Female ducks had higher pH1 and pH24 of thigh muscle on low-protein diets (*p* < 0.0001), whereas they had no difference on high-protein diets (*p* > 0.05). Female ducks exhibited increased b* values in thigh muscles whether fed with high- or low-protein diets (*p* < 0.03). The L* value, a* value, and water loss rate had no difference between female and male, whether fed with high- or low-protein diets (*p* > 0.05).

### 3.3. Chemical Meat Quality

Dietary protein levels had no influence on the percentage of moisture, protein, fat, and collagen in breast muscle and thigh muscle of the slow-growing ducks (*p* > 0.05; Table 4). Male ducks had a higher percentage of protein and a lower percentage of fat in breast muscle and thigh muscle, whether on high- or low-protein diets (*p* < 0.05). The percentage of collagen was higher in male ducks fed with low-protein diets than that in female ducks (*p* < 0.0001). The percentage of moisture had no difference between females and males, whether fed with high- or low-protein diets (*p* > 0.05).

### 3.4. Amino Acid Contents in Breast Muscle

Dietary protein levels had no influence on amino acid contents in the breast muscle (*p* > 0.05, Table 5). The contents of aspartic acid, threonine, serine, glutamic acid, glycine, tyrosine, and arginine of the breast muscle were higher in female ducks fed with high-protein diets than those in male ducks (*p* < 0.05), but there was no difference between female and male ducks fed with low-protein diets (*p* > 0.05). The contents of alanine, valine, methionine, isoleucine, leucine, and lysine of the breast muscle were higher in male ducks fed with low-protein diets than those in female ducks (*p* < 0.05). The male ducks fed with low-protein diets and female ducks fed with high-protein ones had higher phenylalanine content in breast muscle (*p* < 0.05).

### 3.5. Amino Acid Contents in Thigh Muscle

Dietary protein levels had no influence on amino acid contents in the thigh muscle (*p* > 0.05, Table 6). The contents of serine, alanine, and methionine in the thigh muscle were higher in female ducks fed with high-protein diets than those of male ducks (*p* < 0.05), but there was no difference between female and male ducks fed with low-protein diets (*p* > 0.05). Female ducks had higher isoleucine and histidine contents, and lower lysine and arginine contents when fed on low-protein diets (*p* < 0.05). The contents of aspartic acid, threonine, glutamic acid, glycine, valine, leucine, tyrosine, and proline had no difference between female and male ducks, whether fed with high- or low-protein diets (*p* > 0.05).

### 3.6. Fatty Acid Contents in Breast Muscle

Dietary protein levels had no influence on fatty acid contents in the breast muscle (*p* > 0.05, Table 7), but changed the ratio of ω-3/ω-6 (*p* < 0.05). A high-protein diet increased the ratio of ω-3/ω-6 in breast muscle (*p* < 0.05). Male ducks had higher contents of C16:0,

C18:1, C18:2, C18:3, SFA, MUFA, PUFA, ω-3, and ω-6, whether on high- or low-protein diets (*p* < 0.05). Male ducks fed with low-protein diets had higher C16:0 content in breast muscle (*p* < 0.05), and female ducks fed with a low-protein diet had lower C16:1 and C17:0 contents (*p* < 0.05).

### 3.7. Fatty Acid Contents in Thigh Muscle

It was observed that high-protein diets increased the contents of C14:0, C16:0, C16:1, C18:1, C20:4, SFA, MUFA, and ω-6 fatty acids (*p* < 0.05, Table 8), and reduced the contents of C22:6, ω-3 fatty acids, and ω-3/ω-6 ratio in thigh muscle (*p* < 0.05). The fatty acids contents in thigh muscle had no differences between female and male ducks, whether fed with high- or low-protein diets (*p* > 0.05).

## 4. Discussion

Several attempts have been made to reduce the crude protein content in duck diets. Researchers generally agree that reduced dietary crude protein adversely affects growth performance and appetite [21]. For instance, Pekin ducks fed a low-protein diet had a lower body weight and average weight gain than those fed a high-protein diet [14,22]. However, some studies suggest that the growth of Pekin ducks can be maintained if dietary protein is reduced to a minimal level while ensuring an adequate balance of essential amino acids [23,24]. A study demonstrated that Pekin ducks fed diets containing 13% protein and a balanced ratio of eight amino acids exhibited average weight gains comparable to those in ducks fed 18% protein diets [25]. In the current study, slow-growing ducks experience a decrease in live body weight at 63 days of age when fed a low-protein diet for 42 days. The reason might be that low-protein diets have lower amino acid content than high-protein diets.

As expected, the proportion of breast and thigh muscles did not decrease in slow-growing ducks fed the low-protein diet in the current research. This observation aligns with findings reported on Pekin ducks [25]. However, the literature presents divergent outcomes. Pekin ducks reported that while low-protein diets reduced breast meat yield, they improved carcass and leg meat yields [14]. Additionally, elevated abdominal fat deposition has been documented in Pekin ducks consuming low-protein diets [23]. This might be attributed to an imbalance of amino acids in low-protein diets [25]. The researchers highlighted that this imbalance could result in increased amino acid breakdown, causing carbon skeletons to be redirected toward intermediates for the synthesis of carbohydrates and lipids. Furthermore, a higher calorie-to-protein ratio in low-protein diets results in the accumulation of excess energy beyond the requirements for protein synthesis and deposition, which is subsequently converted to fat and stored as abdominal fat [22]. In contrast to these reports, slow-growing ducks in the current trial did not show increased abdominal fat deposition. This discrepancy may be attributed to the distinct energy partitioning strategy of slow-growing breeds. Unlike fast-growing animals selected for rapid protein deposition and concomitant high energy intake, slow-growing ducks likely prioritize the utilization of limited energy and amino acids for maintenance and baseline protein synthesis over excessive lipogenesis when dietary protein is restricted. This highlights the importance of considering genotype-specific metabolic responses when formulating diets.

Meat quality is a key concern for both producers and consumers. The current study showed that there were no notable variations in the pH_1_ and pH_24_ values of the breast and thigh muscle between ducks fed high- and low-protein diets. By contrast, Pekin ducks fed low-protein diets had a high pH [14]. The ultimate pH is a critical determinant of meat quality, as it directly affects water-holding capacity, texture, and flesh color [26]. A lower pH is generally linked to reduced water-holding capacity, which can increase cooking loss and drip loss, raise shear force, and decrease tenderness and shelf life [27]. Dietary protein does not affect the water-holding capacity of the breast and thigh muscles, but high-protein diets increase the b* value in the present study. This suggests that reducing protein intake does not have an adverse effect on meat quality in slow-growing ducks. Furthermore, female ducks exhibited higher pH_1_; and pH_24_ in both breast and thigh muscles, as well as a higher b* value in thigh muscle compared to males. A finding stated that sex had no influence on the pH_24_ of the breast and thigh muscles in Pekin ducks [6], highlighting potential breed-specific or experimental variations.

The key characteristic influencing the palatability of cooked meat is flavor, which is influenced by several factors, including breed, species, sex, slaughter age, stress levels, muscle type, intramuscular fat content, and animal diet [28,29]. Additionally, the flavor is further modulated by its fatty acid profile [30]. Specifically, the proportions of C18:0, SFA, PUFA, MUFA, and the UFA/SFA ratio have been correlated with flavor intensity [6,30]. Moreover, the levels of SFA and PUFA, PUFA/SFA ratio, and ω-6/ω-3 ratio are commonly used indices for evaluating the nutritional value of dietary foods [31]. Importantly, higher intake of PUFAs, especially ω-3 PUFAs, is associated with a reduced risk of cardiovascular disease [31]. In the current research, a low-protein diet increased the contents of C18:0, C20:3, C22:0, SFA, and MUFA in the breast and thigh muscles of slow-growing ducks and decreased the contents of C18:3 and ω-3 PUFA in the breast muscle, as well as ω-3 PUFA in the thigh muscle. Notably, the muscle-specific response to dietary protein level, observed only in the thigh muscle, can be explained by fundamental differences in muscle fiber composition and metabolic activity. Poultry thigh muscles are predominantly composed of oxidative (Type I and IIa) fibers, which possess greater mitochondrial density and higher β-oxidation capacity, making their lipid metabolism more dynamic and responsive to dietary alterations. In contrast, breast muscle is mainly glycolytic (Type IIb), with energy metabolism more reliant on carbohydrates [32], which likely accounts for its distinct fatty acid modulation under varying dietary protein levels.

The fatty acid profile of meat is consistently influenced by dietary protein levels across species, as evidenced by studies in pigs. Research indicates that a low-protein diet decreases the contents of C16:1, C18:0, C18:3, SFA, and ω-3 PUFA, and increases C18:1 contents in the longissimus lumborum muscle of pigs [33]. Similarly, in the subcutaneous fat of pigs, a reduction in dietary protein (from 20.9% to 18.1%) led to increased saturated and monounsaturated fat and a decrease in polyunsaturated fat [34]. A more comprehensive study reported that a low-protein diet reduced the contents of C18:2, C18:3, C20:2, C20:4, C20:5, C24:1, PUFA, and ω-3 and ω-6 PUFA and increased the contents of C14:0, C16:0, C18:0, C20:0, SFA, and MUFA in the longissimus thoracis muscle of finishing barrows [35]. Concurrently, reductions in C18:1, C18:3, C20:2, C24:1, and *ω*-3 PUFA were observed in the semimembranosus muscle. These results indicated that animals fed a low-protein diet exhibited elevated levels of SFA and MUFA and reduced levels of PUFA. Although the low-protein diet increased the nutritional value of meat, it might also increase the risk of developing cardiovascular disease.

Sex exerted a significant main effect on the proportions of individual fatty acids. in the current research, male ducks exhibited elevated levels of C14:0, C15:0, C16:0, C16:1, C17:0, C18:1, C18:2, C20:0, C18:3, C22:1, SFA, MUFA, PUFA, ω-3, and ω-6 PUFA in breast muscles compared to female ducks. This finding is consistent with the report that males had high proportions of C17:0, C18:0, C18:2 ω-6, C18:3 n-3, C20:5 ω-3, C22:6 n-3, PUFA, ω-6 fatty acids, ω-3 fatty acids, distilled fatty acids, and a ratio of PUFA/SFA and low proportions of C18:1, MUFA, and ω-6/ω-3 in the breast muscle [6]. Another study showed high proportions of C12:0, C15:0, C18:0, C18:3, C22:2, SFA, PUFA, PUFA/SFA, UFA/SFA, ω-6, and ω-3 PUFA in the longissimus thoracis muscle samples of male goat kids [36]. These results indicate that breast meat from male ducks had high proportions of SFA, MUFA, and PUFA. This distinct fatty acid profile suggests that meat from males may possess enhanced flavor potential and a modified nutritional value.

Amino acids play a critical role in determining meat quality [37]. The composition and concentration of amino acids in muscle are the primary determinants of the nutritional value and flavor of animal meat, serving as primary indicators for overall meat quality assessment [38]. In particular, taste-related amino acids significantly influence meat palatability and are widely regarded as major flavor enhancers [9,39]. Among these, certain amino acids contribute distinct taste profiles: arginine, leucine, valine, isoleucine, methionine, phenylalanine, and histidine are associated with bitterness, while glutamic acid and aspartic acid impart a savory, refreshing umami taste. Notably, glutamic acid level is a decisive factor in meat flavor perception [40]. In the present study, low-protein diets increased the contents of isoleucine and histidine and reduced the lysine and arginine contents of thigh muscle in female ducks. This pattern may result from a combination of factors: (1) a potential reduction in whole-body protein turnover rate, conserving amino acids within the muscle pool; (2) a preferential redistribution of limited amino acids towards muscle tissue; and (3) a downregulation of catabolic pathways (e.g., transamination or oxidation) for specific amino acids in muscle, leading to their relative accumulation. These adaptive mechanisms could maintain muscle amino acid concentrations despite reduced dietary input, reflecting the metabolic priority of preserving critical tissue function [41]. This suggests that protein intake can affect the meat quality with gender-dependent effects.

The pronounced sex-dependent variations in amino acid and fatty acid profiles observed in this study likely originate from fundamental physiological and metabolic divergences between male and female ducks. Even under identical rearing management and nutritional conditions, male and female poultry exhibit significant differences in growth patterns, final body weight, muscle yield, and body fat percentage. This phenomenon is known as sexual dimorphism [42]. Male broilers typically grow faster, produce more breast meat, and have lower body fat, while females tend to deposit more fat as they approach sexual maturity [43]. Firstly, sex hormones exert profound regulatory effects. Estrogen in females is known to influence lipid metabolism, potentially favoring lipid deposition and altering fatty acid desaturation patterns, whereas androgens in males promote protein anabolism, leading to distinct patterns of amino acid uptake and utilization in muscle tissue [44]. Secondly, inherent metabolic priorities differ; male ducks typically exhibit faster growth rates and higher protein turnover, making their metabolic pathways, particularly for amino acids, more sensitive and responsive to variations in dietary protein levels compared to females [45]. Collectively, the interplay of hormonal regulation, divergent growth and metabolic set points, and possibly underlying muscle physiology provides a coherent mechanistic framework for the strong sex effects documented in our results.

## 5. Conclusions

Low-protein diets increased the fatty acid and amino acid contents of breast muscle in male ducks, indicating that such diets could enhance the flavor of breast muscle with gender-dependent effects in slow-growing ducks. Male ducks fed with low-protein diets had high proportions of SFA, MUFA, and PUFA in breast muscles. Additional research is warranted to clarify the molecular processes responsible for the increase in fatty and amino acids in the breast muscles of slow-growing ducks fed low-protein diets.

## Figures and Tables

**Table 1 animals-16-00079-t001:** Feed formulation and nutrient levels (%).

Ingredients	High-Protein Diet	Low-Protein Diet	Nutrient Levels	High-Protein Diet	Low-Protein Diet
Corn	15.00	15.00	Metabolizable energy ^2^ (MJ/kg)	11.72	11.80
Wheat	22.10	38.20	Crude protein ^3^	18.88	16.14
Wheat Middlings	30.00	4.00	Calcium ^2^	0.93	0.91
Rice Power	5.00	17.70	Total phosphorus ^2^	0.78	0.70
Soybean Meal	12.80	5.00	Methionine ^3^	0.31	0.21
Malt Meal	2.00	2.90	Lysine ^3^	0.72	0.76
Citric Acid Residue	7.00	7.00			
Limestone	1.80	1.90			
Calcium Bisphosphate	0.80	0.60			
Bentonite	2.50	4.8			
Soybean Oil		1.90			
Premix ^1^	1	1			
Total	100	100			

^1^ Vitamins, minerals and nutrients supplied per kilogram of total diet are the following: Cu (CuSO_4_·5H_2_O), 10 mg; Fe (FeSO_4_·7H_2_O), 60 mg; Zn (ZnO), 60 mg; Mn (MnSO_4_·H_2_O), 80 mg; Se (NaSeO_3_), 0.3 mg; I (KI), 0.2 mg; choline chloride, 750 mg; vitamin A (retinyl acetate), 8000 IU; vitamin D3 (Cholcalciferol), 3000 IU; vitamin E (DL-α-tocopheryl acetate), 20 IU; vitamin K3 (menadione sodium bisulphate), 2 mg; thiamin (thiamin mononitrate), 1.5 mg; riboflavin, 4 mg; pyridoxine hydrochloride, 3 mg; cobalamin, 0.02 mg; calcium-D-pantothenate, 10 mg; nicotinic acid, 50 mg; folic acid, 1 mg; and biotin, 0.15 mg. ^2^ These values are as formulated. ^3^ These values were determined by analysis based on triplicate determinations.

**Table 2 animals-16-00079-t002:** The effects of dietary protein levels and gender on the carcass traits of slow-growing ducks (%).

Items	Gender	Live Weight	Carcass Percentage	Half-Dressed Carcass Percentage	Eviscerated Percentage	Leg Muscle Percentage	Breast Muscle Percentage	Abdominal Fat Percentage
Low-Protein	Female	1581 ^b^	88.53	82.26	75.92 ^a^	11.06	14.28	0.77
Diet	Male	1735 ^b^	85.32	79.33	72.21 ^b^	10.75	13.06	0.80
High-Protein	Female	1731 ^b^	85.86	80.28	73.77 ^ab^	10.22	13.57	1.02
Diet	Male	1981 ^a^	85.80	78.54	72.08 ^b^	11.01	13.85	0.52
		59	0.91	0.99	1.02	0.48	0.45	0.21
*p*-value								
	Protein	0.3095	0.5662	0.4729	0.5946	0.5609	0.9550	0.9636
	Gender (Protein)	0.0064	0.0592	0.0777	0.0358	0.4715	0.1643	0.2173

^a,b^ Means within column with different superscripts differ (*p* < 0.05).

**Table 3 animals-16-00079-t003:** The effects of dietary protein levels and gender on the meat quality of breast and thigh muscle in slow-growing ducks.

		Breast Muscle						Thigh Muscle					
Items	Gender	pH_1_	pH_24_	L* ^1^	a* ^1^	b* ^1^	Water Loss Rate, % ^2^	pH_1_	pH_24_	L* ^1^	a* ^1^	b* ^1^	Water Loss Rate, % ^2^
Low-Protein	Female	6.05 ^b^	6.05 ^a^	41.98	12.82	7.17	37.71	6.27 ^a^	6.24^a^	40.66	10.71	9.69 ^a^	38.33
Diet	Male	6.04 ^b^	6.00 ^b^	41.43	12.32	7.56	35.66	6.16 ^b^	6.14^b^	45.00	10.39	7.81 ^b^	38.51
High-Protein	Female	6.11 ^a^	6.04 ^a^	42.1	9.6	8.6	41.71	6.18 ^b^	6.16^b^	42.13	10.61	9.31 ^a^	39.04
Diet	Male	6.05 ^b^	6.02 ^ab^	42.2	7.35	7.65	38.1	6.17 ^b^	6.15^b^	42.20	7.68	7.65 ^b^	36.68
	Pooled SEM	0.012	0.012	1.43	0.90	0.38	2.20	0.010	0.011	1.27	1.33	0.60	1.35
*p*-value													
	Protein	0.3622	0.8008	0.2382	0.0715	0.2817	0.2576	0.5538	0.5337	0.7884	0.4410	0.8502	0.6799
	Gender (Protein)	0.0120	0.0322	0.9634	0.2122	0.1362	0.4269	<0.0001	<0.0001	0.0706	0.3061	0.0218	0.4723

^a,b^ Means within column with different superscripts differ (*p* < 0.05). ^1^ Muscle lightness (L*), redness (a*), and yellowness (b*), as determined by the Commission Internationale de l’Eclairage, were determined using a chroma meter (CR-400, Konica Minolta, Tokyo, Japan). The data were expressed according to CIELAB coordinates, that are, L* = 0 (black) to L* = 100 (white) represents luminosity, and −a* (green), +a* (red), −b* (blue), and +b* (yellow). The surfaces of all the samples were freshly trimmed and free of fat or connective tissue. ^2^ The water loss rate was measured using a meat quality pressure meter (Meat-1, Tenovo Food, Beijing, China), and the calculation method of water loss ratio = weight of meat after pressing/weight of meat before pressing.

**Table 4 animals-16-00079-t004:** The effects of dietary protein levels and gender on the chemical composition of breast and thigh muscle in slow-growing ducks.

		Breast Muscle				Thigh Muscle			
Items	Gender	Moisture ^1^	Protein ^1^	Fat ^1^	Collagen ^1^	Moisture ^1^	Protein ^1^	Fat ^1^	Collagen ^1^
Low-Protein	Female	74.41	22.68 ^b^	2.40 ^a^	1.14	73.59	22.40 ^ab^	3.48 ^a^	1.17 ^b^
Diet	Male	74.41	23.09 ^a^	1.35 ^b^	1.14	73.76	22.90 ^a^	2.75 ^b^	1.33 ^a^
High-Protein	Female	74.88	22.41 ^b^	2.48 ^a^	1.16	73.66	22.28 ^b^	3.41 ^a^	1.18 ^b^
Diet	Male	74.64	23.36 ^a^	1.46 ^b^	1.14	73.94	22.72 ^ab^	2.70 ^b^	1.23 ^b^
	Pooled SEM	0.1367	0.1191	0.0443	0.0099	0.1293	0.1729	0.0633	0.0211
*p*-value									
	Protein	0.0978	0.9940	0.9106	0.5664	0.5207	0.6972	0.9217	0.6217
	Gender (Protein)	0.4618	<0.0001	<0.0001	0.2507	0.2141	0.0367	<0.0001	<0.0001

^a,b^ Means within column with different superscripts differ (*p* < 0.05). ^1^ It was detected and generated by the FoodScan (FOSS FoodScan 78800, Dedicated Analytical Solutions, Hilleroed, Denmark) instrument, and was usually expressed as a percentage (%), representing the proportion of the nutrient in the sample relative to its total mass (on a wet weight basis).

**Table 5 animals-16-00079-t005:** The effects of dietary protein levels and gender on the amino acid composition of breast muscle in slow-growing ducks (g/100 g) ^1^.

Items	Gender	Aspartic Acid	Threonine	Serine	Glutamic Acid	Glycine	Alanine	Cystine	Valine	Methionine	Isoleucine	Leucine	Tyrosine	Phenylalanine	Lysine	Histidine	Arginine	Proline
Low-Protein	Female	1.92 ^a^	0.93 ^a^	0.77 ^a^	3.20 ^a^	0.91 ^a^	1.22 ^bc^	0.12	0.98 ^b^	0.55 ^b^	0.99 ^b^	1.69 ^b^	0.86 ^a^	0.97 ^b^	1.99 ^b^	0.63 ^b^	1.43 ^a^	0.74
Diet	Male	1.92 ^a^	0.90 ^a^	0.75 ^a^	3.25 ^a^	0.95 ^a^	1.41 ^a^	0.18	1.05 ^a^	0.59 ^a^	1.03 ^a^	1.85 ^a^	0.93 ^a^	1.05 ^a^	2.07 ^a^	0.69 ^a^	1.48 ^a^	0.81
High-Protein	Female	1.91 ^a^	0.92 ^a^	0.75 ^a^	3.24 ^a^	0.93 ^a^	1.30 ^b^	0.17	0.98 ^b^	0.57 ^ab^	0.99 ^b^	1.72 ^b^	0.85^a^	1.03 ^a^	1.96 ^b^	0.66 ^ab^	1.48 ^a^	0.71
Diet	Male	1.79 ^b^	0.87 ^b^	0.71 ^b^	3.04 ^b^	0.82 ^b^	1.18 ^c^	0.18	0.96 ^b^	0.56 ^b^	0.92 ^c^	1.63 ^b^	0.71 ^b^	0.94 ^b^	1.85 ^c^	0.66 ^ab^	1.37 ^b^	0.70
	Pooled SEM	0.02	0.01	0.009	0.04	0.02	0.03	0.01	0.02	0.1	0.01	0.03	0.02	0.02	0.02	0.02	0.02	0.04
*p*-value																		
	Protein	0.3495	0.5429	0.3599	0.4928	0.4560	0.5569	0.4431	0.3590	0.7192	0.3487	0.4278	0.3016	0.7119	0.2243	0.9296	0.6547	0.1784
	Gender (Protein)	0.0002	0.0028	0.0020	0.0035	0.0024	0.0001	0.4063	0.0065	0.0314	0.0014	0.0019	0.0010	0.0009	0.0004	0.0348	0.0026	0.4963

^a–c^ Means within column with different superscripts differ (*p* < 0.05). ^1^ The content is expressed as grams of each amino acid per 100 g of duck breast muscle on a wet weight basis.

**Table 6 animals-16-00079-t006:** The effects of dietary protein levels and gender on the amino acid content of thigh muscle in slow-growing ducks (g/100 g) ^1^.

Items	Gender	Aspartic Acid	Threonine	Serine	Glutamic Acid	Glycine	Alanine	Cystine	Valine	Methionine	Isoleucine	Leucine	Tyrosine	Phenylalanine	Lysine	Histidine	Arginine	Proline
Low-Protein	Female	1.91	0.89	0.70 ^b^	3.19	0.94	1.21 ^ab^	0.17 ^c^	0.96	0.55 ^b^	1.05 ^a^	1.70	0.83	0.98	2.05 ^a^	0.69 ^a^	1.49 ^a^	0.75
Diet	Male	1.80	0.88	0.70 ^b^	3.11	0.88	1.11 ^b^	0.23 ^a^	0.97	0.56 ^ab^	0.93 ^b^	1.55	0.82	0.92	1.87 ^b^	0.61 ^b^	1.35 ^b^	0.76
High-Protein	Female	1.91	0.90	0.70 ^b^	3.12	0.90	1.12 ^b^	0.17 ^c^	0.95	0.53 ^b^	0.96 ^b^	1.60	0.79	0.90	2.02 ^a^	0.61 ^b^	1.48 ^a^	0.71
Diet	Male	1.88	0.91	0.74 ^a^	3.21	0.87	1.26 ^a^	0.21 ^b^	1.00	0.60 ^a^	0.98 ^b^	1.63	0.87	0.96	1.97 ^a^	0.63 ^ab^	1.44 ^a^	0.76
*p*-value																		
	Protein	0.5322	0.1128	0.5143	0.8404	0.4570	0.7674	0.7784	0.7536	0.7306	0.7746	0.9019	0.8807	0.6605	0.7361	0.5048	0.6101	0.5384
	Gender (Protein)	0.0667	0.6907	0.0468	0.2262	0.0655	0.0125	0.0025	0.2586	0.0382	0.0018	0.7030	0.4807	0.1988	0.0040	0.0333	0.0052	0.5311

^a–c^ Means within column with different superscripts differ (*p* < 0.05). ^1^ The content is expressed as grams of each amino acid per 100 g of duck breast muscle on a wet weight basis.

**Table 7 animals-16-00079-t007:** The effects of dietary protein levels and gender on the fatty acid composition of breast muscle in slow-growing ducks (g/10,000 g) ^1^.

Items	Gender	C14:0	C15:0	C16:0	C16:1	C17:0	C18:0	C18:1	C18:2	C20:0	C18:3	C20:1	C22:1	C22:0	C20:3	C22:1	C20:4	C23:0	C20:5	C24:0	C24:1	C22:6	SFA ^2^	MUFA	PUFA	ω-3	ω-6	ω-3/ω-6
Low-Protein	Female	0.38 ^b^	0.13	15.58 ^b^	0.80 ^b^	0.42 ^c^	22.70	23.17b	30.92 ^b^	0.72	0.71 ^c^	0.36	1.03	0.64	0.85	26.29	0.44	0.18	0.16	0.39	0.25	0.78	41.16 ^b^	50.88 ^b^	34.90 ^b^	1.63 ^c^	32.25 ^b^	0.0502
Diet	Male	0.52 ^a^	0.15	19.46 ^a^	1.09 ^a^	0.52 ^ab^	24.76	30.66 ^a^	39.55 ^a^	0.76	0.92 ^bc^	0.37	1.34	0.58	1.01	28.63	0.50	0.19	0.21	0.40	0.28	0.95	47.22 ^a^	61.03 ^a^	44.68 ^a^	2.04 ^b^	41.31 ^a^	0.0495
High-Protein	Female	0.36 ^b^	0.12	15.45 ^b^	0.89 ^ab^	0.47 ^bc^	19.43	23.39 ^b^	33.12 ^b^	0.71	1.07 ^b^	0.37	0.66	0.52	0.50	22.70	0.35	0.20	0.16	0.35	0.18	0.93	37.60 ^c^	47.53 ^b^	36.83b	2.15 ^ab^	34.03 ^b^	0.0633
Diet	Male	0.41 ^b^	0.16	17.74 ^a^	1.09 ^a^	0.59 ^a^	20.65	26.63 ^ab^	39.17 ^a^	0.75	1.44 ^a^	0.43	0.86	0.54	0.60	23.28	0.33	0.24	0.17	0.38	0.23	0.95	41.48 ^b^	51.66 ^b^	43.62 ^a^	2.56 ^a^	40.20 ^a^	0.0634
	Pooled SEM	0.03	0.01	0.66	0.09	0.03	1.08	1.37	2.01	0.05	0.11	0.04	0.06	0.07	0.10	1.65	0.06	0.02	0.03	0.02	0.02	0.08	1.78	2.45	2.18	0.15	2.02	0.0022
Protein	Low-Protein Diet	0.45	0.14	17.52	0.94	0.47	23.73	26.92	35.23	0.74	0.81	0.36	1.18	0.61	0.93	27.46	0.47	0.19	0.19	0.39	0.27	0.87	44.19	55.95	39.78	1.83	36.78	0.0498 b
	High-Protein Diet	0.38	0.14	16.59	0.99	0.53	20.04	25.01	36.15	0.73	1.25	0.40	0.76	0.53	0.55	22.99	0.34	0.22	0.16	0.36	0.20	0.94	39.54	49.59	40.22	2.36	37.11	0.0634 a
	Pooled SEM	0.02	0.01	0.47	0.06	0.02	0.77	0.97	1.42	0.03	0.08	0.03	0.04	0.05	0.07	1.17	0.04	0.01	0.02	0.02	0.02	0.06	1.26	1.73	1.54	0.11	1.43	0.0016
*p*-value																												
	Protein	0.4820	0.8210	0.7193	0.8175	0.4917	0.0917	0.6866	0.8791	0.7810	0.1729	0.3399	0.4448	0.1417	0.0657	0.0541	0.4514	0.2433	0.5483	0.1578	0.2402	0.4922	0.3256	0.3659	0.9482	0.2142	0.9567	0.0006
	Gender (Protein)	0.0125	0.0512	0.0004	0.0358	0.0036	0.3144	0.0018	0.0053	0.6893	0.0363	0.5156	0.2988	0.7632	0.5952	0.7821	0.2963	0.3590	0.4459	0.7158	0.1314	0.3462	0.0356	0.0172	0.0038	0.0403	0.0041	0.9774

^1^ The content is expressed as grams of each amino acid per 10,000 g of duck breast muscle on a wet weight basis. ^2^ Saturated fatty acids, SFA; Monounsaturated fatty acids, MUFA; Polyunsaturated fatty acids, PUFA. ^a–c^ Means within column with different superscripts differ (*p* < 0.05).

**Table 8 animals-16-00079-t008:** The effects of dietary protein levels and gender on the fatty acid composition of thigh muscle in slow-growing ducks (g/10,000 g) ^1^.

Items	Gender	C14:0	C15:0	C16:0	C16:1	C17:0	C18:0	C18:1	C18:2	C20:0	C18:3	C20:1	C22:1	C22:0	C20:3	C22:1	C20:4	C24:0	C24:1	C22:6	SFA ^2^	MUFA	PUFA	ω-3	ω-6	ω-3/ω-6
Low-Protein	Female	1.03	0.24	27.5	2.92	0.69	30.06	88.17	74.80	0.62	2.13	1.03	0.37	0.62	20.50	0.33	0.42	0.28	0.21	0.94	61.30	112.83	79.03	3.08	75.95	0.0416
Diet	Male	1.15	0.27	29.6	3.25	0.65	30.31	95.77	80.37	0.66	2.33	1.18	0.33	0.66	20.35	0.30	0.42	0.34	0.18	0.95	63.79	120.73	85.18	3.30	81.51	0.0429
High-Protein	Female	0.56	0.19	20.1	2.31	0.58	23.40	53.93	58.23	0.62	2.56	0.84	0.36	0.35	20.34	0.20	0.17	0.30	0.16	1.26	46.32	77.57	63.61	3.91	58.95	0.0677
Diet	Male	0.61	0.21	21.78	2.32	0.61	25.52	58.87	63.73	0.67	2.70	0.88	0.36	0.38	21.85	0.23	0.17	0.31	0.17	1.29	50.27	84.09	69.45	4.21	64.50	0.0678
	Pooled SEM	0.14	0.034	3.86	0.56	0.068	1.42	14.09	11.48	0.044	0.46	0.17	0.032	0.051	0.85	0.021	0.016	0.037	0.026	0.10	5.45	14.35	11.96	0.41	11.55	0.0034
Protein	Low-Protein Diet	1.09 ^a^	0.26	28.55 ^a^	3.09 ^a^	0.67	30.19 ^a^	91.97 ^a^	77.59	0.64	2.22	1.11	0.35	0.64 ^a^	20.43	0.31 ^a^	0.42 ^a^	0.31	0.19	0.94 ^b^	62.54 ^a^	116.78 ^a^	82.10	3.19 ^b^	78.73 ^a^	0.0422 ^b^
	High-Protein Diet	0.58 ^b^	0.20	20.94 ^b^	2.31 ^b^	0.59	24.46 ^b^	56.40 ^b^	60.98	0.64	2.63	0.86	0.36	0.36 ^b^	21.10	0.22 ^b^	0.17 ^b^	0.31	0.16	1.27 ^a^	48.30 ^b^	80.83 ^b^	66.53	4.06 ^a^	61.72 ^b^	0.0677 ^a^
	Pooled SEM	0.099	0.024	2.74	0.40	0.048	1.01	9.96	8.12	0.031	0.33	0.12	0.022	0.036	0.60	0.014	0.011	0.026	0.018	0.07	3.85	10.15	8.46	0.29	8.17	0.0024
*p*-value																										
	Protein	0.0143	0.0767	0.0298	0.0438	0.0899	0.0330	0.0158	0.0513	0.9617	0.0798	0.0823	0.6940	0.0085	0.4720	0.0512	<0.0001	0.5473	0.1198	0.0028	0.0258	0.0197	0.0668	0.0443	0.0494	0.0007
	Gender (Protein)	0.8265	0.7733	0.8874	0.9156	0.8851	0.5787	0.9023	0.8910	0.6152	0.9338	0.8283	0.7195	0.7808	0.4595	0.2955	0.9974	0.8780	0.7921	0.9701	0.8339	0.8812	0.8826	0.8142	0.8914	0.9581

^1^ The content is expressed as grams of each amino acid per 10,000 g of duck breast muscle on a wet weight basis. ^2^ Saturated fatty acids, SFA; Monounsaturated fatty acids, MUFA; Polyunsaturated fatty acids, PUFA. ^a,b^ Means within column with different superscripts differ (*p* < 0.05).

## Data Availability

Data presented in this study are available upon request from the corresponding author.

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
