# Peer review of "Impact of Dietary Protein Levels and Gender on Carcass Characteristics and Meat Quality in Slow-Growing Ducks"

_animals, 2025, doi:10.3390/ani16010079_

Round 1
Reviewer 1 Report
Comments and Suggestions for Authors
Hello dear,
I am submitting my comments on manuscript “Impact of dietary protein levels and gender on carcass characteristics and meat quality in slow-growing ducks”. The following points are provided for your review and consideration.
1) Lack of mechanistic explanation for sex-driven differences. The manuscript repeatedly highlights strong sex effects on amino acid and fatty acid profiles, but no biological mechanisms (hormonal, metabolic, or muscle-fiber differences) are discussed. A mechanistic interpretation is needed to justify these gender-dependent responses.
2) Unexpected absence of abdominal fat increase in low-protein diets. Most literature reports elevated abdominal fat under low-protein diets, yet this study shows no such response. The discussion should clarify why slow-growing ducks deviate from this well-established pattern and propose physiological explanations.
3) No explanation for why fatty acid changes appear only in thigh muscle Protein level affected fatty acids in thigh but not breast muscle. The discussion should address possible causes, such as differences in oxidative capacity, fiber composition, or intramuscular fat deposition between the two muscles.
4) Inconsistent amino acid patterns under low-protein diets. The increase of certain amino acids in muscle despite dietary protein reduction contradicts expected metabolic responses. A brief explanation regarding amino acid redistribution, catabolism, or altered utilization would strengthen the interpretation.
5) Missing mechanism behind increased SFA and MUFA under low protein. The manuscript reports higher SFA and MUFA under low-protein diets but does not explore underlying metabolic pathways. A mechanistic rationale is necessary to support the conclusion.
Best regards,
Author Response
Dear Editor,
Thanks so much for your email dated on 26 November 2025 for our manuscript (animals-4012198) for revisions. We had revised manuscript [R1] according to the reviewers’ comments and recommendations and also submitted the revised manuscript [R1] to the journal site (https://mc04.manuscriptcentral.com/ps).we had revised the manuscript thoroughly, and the revisions are highlighted in full text by using red colour. All Authors have read the revised manuscript [R1] and have agreed to submit it in its current form for consideration for publication in Animals. We would like to answer your further questions or do further revisions if these answers or revisions are not clear or if you need additional information. Your help and support would be highly appreciated.
Best regards,
Yong Jiang, PhD,
Response to Reviewer 1:
I am submitting my comments on manuscript “Impact of dietary protein levels and gender on carcass characteristics and meat quality in slow-growing ducks”. The following points are provided for your review and consideration.
Comments 1: Lack of mechanistic explanation for sex-driven differences. The manuscript repeatedly highlights strong sex effects on amino acid and fatty acid profiles, but no biological mechanisms (hormonal, metabolic, or muscle-fiber differences) are discussed. A mechanistic interpretation is needed to justify these gender-dependent responses.
Response 1: We had revised this manuscript according to your suggestions in L390-407.
Comments 2: Unexpected absence of abdominal fat increase in low-protein diets. Most literature reports elevated abdominal fat under low-protein diets, yet this study shows no such response. The discussion should clarify why slow-growing ducks deviate from this well-established pattern and propose physiological explanations.
Response 2: We had revised this manuscript according to your suggestions in L305-311.
Comments 3: No explanation for why fatty acid changes appear only in thigh muscle Protein level affected fatty acids in thigh but not breast muscle. The discussion should address possible causes, such as differences in oxidative capacity, fiber composition, or intramuscular fat deposition between the two muscles.
Response 3: We had revised this manuscript according to your suggestions in L337-344.
Comments 4: Inconsistent amino acid patterns under low-protein diets. The increase of certain amino acids in muscle despite dietary protein reduction contradicts expected metabolic responses. A brief explanation regarding amino acid redistribution, catabolism, or altered utilization would strengthen the interpretation.
Response 4: We had revised this manuscript according to your suggestions in L381-388.
Comments 5: Missing mechanism behind increased SFA and MUFA under low protein. The manuscript reports higher SFA and MUFA under low-protein diets but does not explore underlying metabolic pathways. A mechanistic rationale is necessary to support the conclusion.
Response 5: This study did not involve the measurement of gene or protein expression levels. therefore, we were unable to explore the underlying mechanisms and metabolic pathways responsible for the increase in SFA and MUFA under low protein. In subsequent experiments, we will conduct dedicated research on this mechanism in accordance with your suggestions.

Reviewer 2 Report
Comments and Suggestions for Authors
- In the part of Summary, the last 2 sentences say the same. Should be corrected.
- In the chapter entitled "Abstract," the findings are missing information about what the values increased or decreased in relation to. The results cannot be interpreted on their own.
- If the aim was to ensure that the two experimental treatments differed only in terms of the crude protein content of the feed mixtures, then the calcium and total phosphorus levels should have been set at the same level. It is also unclear why there is such a large difference in the levels of these minerals and what causes it. This is a flaw in the experimental design.
- The description of the Statistics should include two-way ANOVA, as this is evident from the results tables.
- Two data points are included in the evaluation of the pH results, but the Materials and Methods section mentions three occasions for sampling. This needs to be corrected. It should also be noted what exactly the numbers (1 and 24) behind the pH mean.
- The P-values in Tables 7 and 8 were either reversed between the two factors (protein level and gender) or were not evaluated correctly, because the authors consistently wrote that protein level had an effect, not gender, even though the opposite was true based on the P-values. This needs to be corrected based on valid results.
- Lines 283-285: Where can this result be seen? Only Table 2 shows live weight data, but this does not support this claim. The authors also state there that protein level does not affect live weight, only gender has an effect on it.
- Feed conversion ratio data may be interesting because it would show how much longer ducks a low-protein diet had to be kept before they reached the technological slaughter weight.
- Lines 308-309: there is a typo in the sentence, as it contains both the phrases "increased" and "had no significant."
- Lines 332-334: Here too, a correction is needed to ensure that the P-values in Table 7 are correct or that the evaluation is incorrect. The same applies to lines 372-373 (relating to Table 8).
- Lines 372-373: This conclusion does not follow from the previous one.
- The entire conclusion is incorrect based on the results, or the results are reported incorrectly (P values in Tables 7 and 8?).
Author Response
Dear Editor,
Thanks so much for your email dated on 26 November 2025 for our manuscript (animals-4012198) for revisions. We had revised manuscript [R1] according to the reviewers’ comments and recommendations and also submitted the revised manuscript [R1] to the journal site (https://mc04.manuscriptcentral.com/ps).we had revised the manuscript thoroughly, and the revisions are highlighted in full text by using red colour. All Authors have read the revised manuscript [R1] and have agreed to submit it in its current form for consideration for publication in Animals. We would like to answer your further questions or do further revisions if these answers or revisions are not clear or if you need additional information. Your help and support would be highly appreciated.
Best regards,
Yong Jiang, PhD,
Response to Reviewer 2:
I am submitting my comments on manuscript “Impact of dietary protein levels and gender on carcass characteristics and meat quality in slow-growing ducks”. The following points are provided for your review and consideration.
Comments 1: In the part of Summary, the last 2 sentences say the same. Should be corrected.
Response 1: We deleted the second to last sentence.
Comments 2: In the chapter entitled "Abstract," the findings are missing information about what the values increased or decreased in relation to. The results cannot be interpreted on their own.
Response 2: We had We have added the missing information in L33-35.
Comments 3: If the aim was to ensure that the two experimental treatments differed only in terms of the crude protein content of the feed mixtures, then the calcium and total phosphorus levels should have been set at the same level. It is also unclear why there is such a large difference in the levels of these minerals and what causes it. This is a flaw in the experimental design.
Response 3: Based on your feedback, we have checked the feed formula again and found some errors in it. We are very sorry that we did not write this manuscript carefully. We have now corrected information with red colour in Table 1. In future research, we will carefully recheck the data in the manuscript.
Comments 4: The description of the Statistics should include two-way ANOVA, as this is evident from the results tables.
Response 4: We raised both male and female ducks in the same pen. During the slaughter and sampling process, we separated the male and female ducks for sampling and data analysis. Therefore, we believe that the gender should be nested within the pen. We did not use two-factor (protein levels and gender) analysis of variance, but conducted the analysis according to the nested design.
Comments 5: Two data points are included in the evaluation of the pH results, but the Materials and Methods section mentions three occasions for sampling. This needs to be corrected. It should also be noted what exactly the numbers (1 and 24) behind the pH mean.
Response 5: I'm sorry, our previous statement was not clear, and we had corrected it in L123-124. pH1 refers to the pH value measured 1 hour after meat sample collection, and correspondingly, pH24 refers to the pH value measured 24 hours after meat sample collection
Comments 6: The P-values in Tables 7 and 8 were either reversed between the two factors (protein level and gender) or were not evaluated correctly, because the authors consistently wrote that protein level had an effect, not gender, even though the opposite was true based on the P-values. This needs to be corrected based on valid results.
Response 6: It was our mistake. We confused Table 7 and Table 8. It has now been corrected in Table 7 and Table 8.
Comments 7: Lines 283-285: Where can this result be seen? Only Table 2 shows live weight data, but this does not support this claim. The authors also state there that protein level does not affect live weight, only gender has an effect on it.
Response 7: we had checked the data in Table 2, which does not support this claim. Therefore, we deleted this sentence, to avoid any misinterpretation of this manuscript.
Comments 8: Feed conversion ratio data may be interesting because it would show how much longer ducks a low-protein diet had to be kept before they reached the technological slaughter weight.
Response 8: Due to our carelessness, we failed to accurately record the feed intake, so we were unable to calculate the feed conversion ratio. In future research, we will carefully record the data.
Comments 9: Lines 308-309: there is a typo in the sentence, as it contains both the phrases "increased" and "had no significant."
Response 9: It was revised in L312-314.
Comments 10: Lines 332-334: Here too, a correction is needed to ensure that the P-values in Table 7 are correct or that the evaluation is incorrect. The same applies to lines 372-373 (relating to Table 8).
Response 10: I am sorry for our mistake. We confused Table 7 and Table 8, and It has now been corrected.
Comments 11: Lines 372-373: This conclusion does not follow from the previous one.
Response 11: We had revised sentence in L388-389.
Comments 12: The entire conclusion is incorrect based on the results, or the results are reported incorrectly (P values in Tables 7 and 8?).
Response 12: It was our mistake. We confused Table 7 and Table 8. It has now been corrected.

Reviewer 3 Report
Comments and Suggestions for Authors
The study has been thoroughly reviewed.
The abstract is concise and clearly written.
In line 31, days 22-63 correspond to 41 days. The authors should check the accuracy of the sentence.
In the introduction,
In line 56, no supporting information could be found in the literature. The author should check the accuracy of the literature or add other literature supporting the sentence.
In the Materials and Methods section, ask whether the live weights of the ducks in each group and each replicate were equal at the beginning of the trial. This information must be stated in the article. If possible, the average live weights per trial should be included in the performance table.
If performance values ​​were calculated by weighing 100 ducks from each replicate in line 109, why weren't the exact values ​​obtained by weighing 500 ducks? In other words, how do the authors know that the values ​​obtained from 100 birds are equal to the values ​​obtained from 500 ducks?
Correct line 132 cm3.
How were the n3/n6 ratios calculated in Tables 7 and 8? I couldn't understand the values ​​in the table.
In Line 164, the hydrolysis process was carried out for 22 hours at 110 degrees Celsius in what kind of container? Was it a glass bottle with a lid? The bottle specifications should be provided.
Table 1 is incomprehensible. For example, the carcass value is given, but is this the carcass yield? Normally, values ​​of 70 or closer to carcass yield should be obtained. The values ​​given for carcass1, half-dressed carcass1, and eviscerated1 are presumably eviscerated carcass yields. These values ​​should be easier for readers of the article to understand.
As I understand it from the experiment, 6,000 ducks were divided into six replicates of 500 ducks each, and two treatment groups of 3,000 ducks were created. At the end of the experiment, the weights of 50 females and 50 males were taken. All animals should have been weighed to determine the recurrence rates. The authors fail to recognize that the average weight obtained from 100 animals represents the weight of 500 animals. This presents a critical error. The large sample size does not prove scientific accuracy. The authors should have calculated live weight gain by taking the animals' total weights at the beginning of the trial and subtracting the difference from the value obtained at the end of the trial. They should also have determined feed intake and feed conversion rates.
Tables 7 and 8 show the fatty acid values ​​for the two treatment groups, with protein values ​​listed below. I believe the table needs to be corrected. It's unclear.
Line 359 mentions the significant impact of amino acids on meat quality, but reference 36 doesn't contain such information; it only mentions the effects of fatty acids and vitamins on meat quality. The literature should be checked and corrected.
Lines 367-368 also state that glycine, alanine, and serine give meat a sweet taste [40]. A review of the literature in number 40 reveals that the study was not about the taste of meat, but about dry-cured Thai sausage. Either the literature should be corrected or it should be removed.
Author Response
Dear Editor,
Thanks so much for your email dated on 26 November 2025 for our manuscript (animals-4012198) for revisions. We had revised manuscript [R1] according to the reviewers’ comments and recommendations and also submitted the revised manuscript [R1] to the journal site (https://mc04.manuscriptcentral.com/ps).we had revised the manuscript thoroughly, and the revisions are highlighted in full text by using red colour. All Authors have read the revised manuscript [R1] and have agreed to submit it in its current form for consideration for publication in Animals. We would like to answer your further questions or do further revisions if these answers or revisions are not clear or if you need additional information. Your help and support would be highly appreciated.
Best regards,
Yong Jiang, PhD,
Response to Reviewer 3:
I am submitting my comments on manuscript “Impact of dietary protein levels and gender on carcass characteristics and meat quality in slow-growing ducks”. The following points are provided for your review and consideration.
Comments 1: In line 31, days 22-63 correspond to 41 days. The authors should check the accuracy of the sentence.
Response 1: It was revised in L29.
Comments 2: In the introduction, In line 56, no supporting information could be found in the literature. The author should check the accuracy of the literature or add other literature supporting the sentence.
Response 2: It was revised in L54-56.
Comments 3: In the Materials and Methods section, ask whether the live weights of the ducks in each group and each replicate were equal at the beginning of the trial. This information must be stated in the article. If possible, the average live weights per trial should be included in the performance table.
Response 3:The average live weight was about 640g in each group and each replicate at the beginning of the trial. It was added in L93-94.
Comments 4: If performance values were calculated by weighing 100 ducks from each replicate in line 109, why weren't the exact values obtained by weighing 500 ducks? In other words, how do the authors know that the values obtained from 100 birds are equal to the values obtained from 500 ducks?
Response 4: Thank you for raising this important methodological question. We agree that ensuring the representativeness of sample data is central to the experimental design. In our study, measuring all individuals within each replicate (500 ducks) was impractical from an operational standpoint and would have introduced significant stress-related interference. Therefore, we employed a random sampling method widely accepted in livestock research. From each replicate, 100 ducks (50 males and 50 females) were randomly selected. This sample size, representing 20% of the population, is sufficient to provide a statistically powerful and reliable estimate of the average performance of the group.
We acknowledge that there will be sampling error between the sample mean and the true population mean. This is precisely why we employed statistical analyses, such as ANOVA, to assess whether the treatment effects were significantly greater than the random error. Our data analysis revealed significant differences between the various treatments, which supports the validity of our sampling approach.
Comments 5: Correct line 132 cm3.
Response 5: It was revised in L134.
Comments 6: How were the n3/n6 ratios calculated in Tables 7 and 8? I couldn't understand the values in the table.
Response 6: It was revised in table 7 and 8.
Comments 7: In Line 164, the hydrolysis process was carried out for 22 hours at 110 degrees Celsius in what kind of container? Was it a glass bottle with a lid? The bottle specifications should be provided.
Response 7: hydrolysis bottle used in amino acid of feedstuff is a glass bottle with a lid, and the bottle specifications was provided in L164-165.
Comments 8: Table 1 is incomprehensible. For example, the carcass value is given, but is this the carcass yield? Normally, values of 70 or closer to carcass yield should be obtained. The values given for carcass1, half-dressed carcass1, and eviscerated1 are presumably eviscerated carcass yields. These values should be easier for readers of the article to understand.
Response 8: It was revised in Table 2.
Comments 9: As I understand it from the experiment, 6,000 ducks were divided into six replicates of 500 ducks each, and two treatment groups of 3,000 ducks were created. At the end of the experiment, the weights of 50 females and 50 males were taken. All animals should have been weighed to determine the recurrence rates. The authors fail to recognize that the average weight obtained from 100 animals represents the weight of 500 animals. This presents a critical error. The large sample size does not prove scientific accuracy. The authors should have calculated live weight gain by taking the animals' total weights at the beginning of the trial and subtracting the difference from the value obtained at the end of the trial. They should also have determined feed intake and feed conversion rates.
Response 9: We acknowledge your concern that the average weight of 100 birds is not numerically identical to the average weight of all 500 birds in a pen. However, in large-scale poultry nutrition trials, it is standard practice to use randomized sampling to estimate pen performance when full-pen weighing is logistically prohibitive. The purpose is not to obtain the exact mean of the 500 birds but to obtain an unbiased and statistically reliable estimate of it.
Weighing every bird in a large pen (10m × 10m, 500 birds) at the same time point is operationally extremely challenging, causes immense stress (which itself affects weight), and risks injury. Random sampling minimizes disruption while providing a valid snapshot.
A sample size of 100 birds per pen (20%) is large and provides high statistical power to detect treatment effects. The core assumption—valid if sampling is truly random—is that the sample mean is an unbiased estimator of the population mean. The statistical tests (ANOVA) then assess whether differences between treatment groups are larger than the natural variation within groups (which includes this sampling variation). The significant differences we reported indicate that the treatment effect exceeded this background noise.
Due to our carelessness, we failed to accurately record the feed intake, so we were unable to calculate the feed conversion ratio.
Comments 10: Tables 7 and 8 show the fatty acid values for the two treatment groups, with protein values listed below. I believe the table needs to be corrected. It's unclear.
Response 10: (protein values) listed below is the p value in table 7 and 8.
Comments 11: Line 359 mentions the significant impact of amino acids on meat quality, but reference 36 doesn't contain such information; it only mentions the effects of fatty acids and vitamins on meat quality. The literature should be checked and corrected.
Response 11: It was revised in L270.
Comments 12: Lines 367-368 also state that glycine, alanine, and serine give meat a sweet taste [40]. A review of the literature in number 40 reveals that the study was not about the taste of meat, but about dry-cured Thai sausage. Either the literature should be corrected or it should be removed.
Response 12: It was removed.

Round 2
Reviewer 3 Report
Comments and Suggestions for Authors
The article titled "Impact of dietary protein levels and gender on carcass characteristics and meat quality in slow-growing ducks" has been re-evaluated.
Following the re-evaluation:
Necessary corrections were made to the abstract.
Corrections were made to the introduction.
No deficiencies were identified in the materials and methods section.
Necessary corrections were made to the results and discussion.